# LLaVA-Critic-R1: Your Critic Model is Secretly a Strong Policy Model

## Abstract

In vision–language modeling, *critic models* are typically trained to evaluate outputs, assigning scalar scores or pairwise preferences, rather than to generate responses. This separation from *policy models*, which produce the responses, is so entrenched that critics are rarely considered for direct policy use. In this work, we challenge this convention. We propose to reorganize preference-labeled critic datasets into verifiable training signals and perform reinforcement learning directly on a base generative model, producing **LLaVA-Critic-R1**, a multimodal critic trained to optimize preference judgments while retaining full generation ability. Surprisingly, LLaVA-Critic-R1 emerges not only as a top-performing critic but also as a competitive policy model—matching or surpassing specialized reasoning VLMs trained with in-domain data across 26 visual reasoning and understanding benchmarks, with an average gain of +5.7% over its base model (Qwen-2.5-VL-7B). Extending this approach to existing strong reasoning VLMs yields **LLaVA-Critic-R1+**, which further advances policy performance without sacrificing critic quality, achieving a SoTA performance of **71.9** on MMMU at the 7B scale. Finally, we show that the enhanced critic ability benefits inference: applying self-critique at test time yields an average +13.8% improvement on five representative reasoning tasks without additional training. Our results reveal that RL training on critic data can produce a unified model excelling at both evaluation and generation, offering a simple path toward scalable, self-improving multimodal systems.

## 1 Introduction

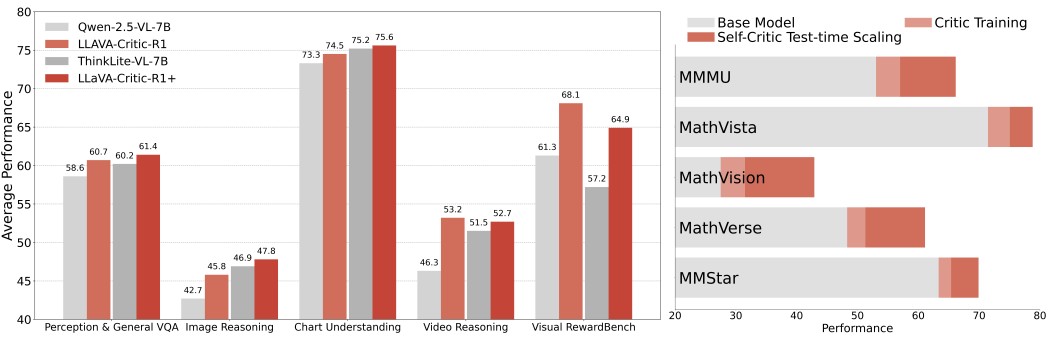

Figure 1: LLaVA-Critic-R1 is trained on top of the base model Qwen-2.5-VL-7B. Building upon a stronger reasoning VLM, ThinkLite-VL-7B, we further develop LLaVA-Critic-R1+ by applying the same RL critic training procedure. **Left:** Performance comparison of LLaVA-Critic-R1 with other base and reasoning VLMs on multiple visual reasoning, visual understanding, and visual reward benchmarks. LLaVA-Critic-R1 not only significantly outperforms other models in critic performance, but also demonstrates stronger policy capabilities. **Right:** Performance improvement of critic training and test-time self-critic scaling on five common visual reasoning and visual understanding benchmarks. Critic training alone significantly improves the base model's performance. Building upon this, leveraging the dual policy and critic capabilities of LLaVA-Critic-R1 for a "Best-of-128" self-critic scaling procedure at test time leads to a further substantial boost in performance.

Vision-language models (VLMs) have achieved impressive results in tasks ranging from image captioning and visual question answering to complex multimodal reasoning (OpenAI, 2023; Liu et al., 2023; Li et al., 2024; Deepmind, 2025; Team et al., 2025a;b). While most attention is paid to *policy models*—systems trained to produce responses—progress has also been driven by *critic models*, which evaluate those responses by assigning scores, ranking pairs, or providing structured feedback. Critic models are central in two stages of the VLM lifecycle: during training, they supply reward signals for reinforcement learning (Wang et al., 2024b; Xiong et al., 2025; Zang et al., 2025), and during inference, they enable test-time scaling by selecting the best candidates from multiple generations (Wang et al., 2024c; Zang et al., 2025; Wang et al., 2025b).

Given this dual functionality, training a powerful critic model is of central importance. In this context, high-quality critic data serves as a foundational component, playing a pivotal role in the effectiveness of critic training. In standard practice, critic datasets consist of a question paired with two responses generated by different models, together with preference judgments provided by human annotators or by more capable reference models. In some cases, they also include pointwise annotations, in which a single model's response is scored by humans or stronger models. These annotated examples are then used to train the critic model through Supervised Fine-Tuning (SFT). More recent approaches augment preference annotations with human- or model-generated rationales, producing *reasoning-rich* critic data (Xiong et al., 2025; Zang et al., 2025), which allows the critic to deliver more informative and interpretable feedback.

In this paper, we explore a novel critic training paradigm enables the critic model to autonomously generate reasoning through reinforcement learning. We begin by reorganizing critic data with preference labels into a verifiable reinforcement learning task. We then perform reinforcement learning directly on a base model to obtain the final critic model, leading to the new model family denoted as LLaVA-Critic-R1.

Surprisingly, LLaVA-Critic-R1 emerges not only as a strong critic but also as a competitive policy model, (1) outperforming its base model by +5.7% on average across 26 diverse visual reasoning and understanding benchmarks, and matching or surpassing specialized reasoning VLMs trained with in-domain data, as shown in Figure 1. (2) Applying our RL critic training to strong reasoning VLMs produces **LLaVA-Critic-R1+**, which further advances policy performance while maintaining top-tier critic ability. (3) Moreover, the improved critic capability enables effective test-time scaling: self-critique boosts performance by +13.8% on five representative reasoning tasks without additional training.

In summary, we highlight the following surprising empirical finding as the key insight of this paper:

> 💡 **Key Findings:**
>
> $(i)$ The critic model trained with RL to judge response preference, can directly be as a strong policy model with improved capabilities generalized to a wide range of downstream tasks.
>
> $(ii)$ This simple approach improves actor/critic capabilities simultaneously in one single model. The improved self-critic capability leads to better test-time scaling behavior.

## 2 CRITIC MODEL TRAINING WITH REINFORCEMENT LEARNING

### 2.1 CRITIC DATA REFORMULATION: FROM SFT TO RL

We begin with the **pairwise** training set of LLaVA-Critic, which contains roughly 40k instances collected from VLFeedback, RLHF, and RLHF-V. Each instance consists of: **(1)** A natural image and an associated question, **(2)** two candidate responses generated by different vision-language models, **(3)** A preference label, provided by either GPT-based annotators or humans, indicating which response is better (or if they are equally good), and **(4)** a long chain-of-thought (Long-CoT) rationale generated by GPT, structured around five explicit evaluation metrics.

While these AI-generated rationales enrich SFT training, we discard them in our RL setting for two reasons: **(a) Avoiding knowledge distillation bias.** Retaining GPT's evaluation criteria risks turning

the model into a passive reproducer of external judgments, rather than an autonomous critic. **(b) Encouraging self-derived reasoning.** Without fixed metric prompts, the model must internally decide how to compare responses, mirroring the open-ended reasoning process of a strong human critic.

Concretely, we strip away the five evaluation metrics and GPT rationales, retaining only the *image*, *question*, and *two responses*. The final RL critic prompt (Table 1) asks the model to decide which response is better (or indicate a tie) and produce the decision in a machine-verifiable format. This ensures that preference labels, whether from humans or GPT, serve as unambiguous ground-truth answers, enabling straightforward reward computation. The resulting formulation is compact, verifiable, and forces the model to *reason for itself*, rather than copy pre-digested justifications.

Table 1: Prompt template used for constructing critic training data.

---

**Prompt Template:**

<image>
You are provided with an image and a question for this image. Please review the corresponding responses and choose which response is better for the given question. Your answer should be the response index if one response is better and "Two responses are equally good." if the responses are equally good. Answer with response number or "Two responses are equally good." directly.
Your task is provided as follows:
Question: {}
Response 1: {}
Response 2: {}

---

## 2.2 INCENTIVIZING CRITIC CAPABILITY VIA REINFORCEMENT LEARNING

We directly conduct reinforcement finetuning without any knowledge distillation SFT. The reward signal $r$ for each data point during training is defined as follows:

$$r = \alpha * r_{\text{pref}} + (1 - \alpha) * r_{\text{format}} \tag{1}$$

$r$ is consisted of the preference reward $r_{\text{pref}}$ and the format reward $r_{\text{format}}$, $\alpha$ is a hyperparameter to balance these two rewards:

- **Preference reward $r_{\text{pref}}$.** The reward is computed based on whether the predicted preference exactly matches the ground truth preference. If yes, the reward is defined as +1, otherwise 0.

- **Format reward $r_{\text{format}}$.** We follow standard practice by encouraging the model to adhere to a specific output format: the critic reasoning process is enclosed within special tokens `<think>...</think>`, and the final answer is wrapped with `\boxed`. The format reward $r_{\text{format}}$ is 1 if it correctly uses the special format tokens and 0 otherwise.

The RL training prompt template is shown in Table 2. We adopt the widely used Group Relative Policy Optimization (GRPO; Shao et al. (2024)) as our training objective, and set $\alpha$ as 0.9 for Eq. 1 during training.

Table 2: Prompt template used for RL training and test-time inference.

---

**Thinking Template:**

You FIRST think about the reasoning process as an internal monologue and then provide the final answer. The reasoning process MUST BE enclosed within <think> </think> tags. The final answer MUST BE put in `\boxed{}`

---

This design delivers two mutually reinforcing benefits: **(1) Enhanced critic capability.** The same model functions as a high-quality evaluator at inference time, reliably selecting the best response from multiple candidates and enabling more effective test-time scaling without additional training. **(2) Policy generalization.** Although post-trained exclusively on critic data, the model acquires transferable reasoning strategies that extend to a wide range of downstream perception, VQA, and

Table 3: General benchmark comparison. All models are prompted with the thinking template, except for the "non-thinking" column of Qwen-2.5-VL-7B. LLaVA-Critic-R1 shows strong critic ability and significantly improves policy performance, matching or surpassing existing reasoning VLMs. LLaVA-Critic-R1+ further enhances policy capability, achieving best results on multiple visual understanding and reasoning tasks. VLM-Agent scores are excluded from overall averages, as some models cannot perform these tasks.

| Benchmark | | Critic Training | | Base Model | | In-domain Policy Training | | | |
| | | LLaVA-Critic-R1 | | Qwen-2.5-VL-7B | | | VL Reasoning Models | | |
| | | (From Base) | + (From Policy) | Non-Thinking | Thinking | ThinkLite-VL-7B | Vision-R1-7B | MM-Eureka-7B | VL-Rethinker-7B |
|---|---|---|---|---|---|---|---|---|---|
| Perception & General VQA | Blink | $58.3_{\uparrow 4.1}$ | $\mathbf{59.0}_{\uparrow 0.5}$ | 54.2 | 53.9 | 58.5 | 56.7 | 54.8 | 56.6 |
| | HallucinationBench$_{Image}$ | $71.3_{\uparrow 3.4}$ | $\mathbf{72.3}_{\uparrow 1.4}$ | 67.9 | 67.2 | 70.9 | 68.8 | 66.9 | 68.3 |
| | MMStar | $65.1_{\uparrow 3.4}$ | $\mathbf{65.7}_{\uparrow 0.7}$ | 61.7 | 62.5 | 65.0 | 60.2 | 64.6 | 65.0 |
| | MMBench | $80.9_{\uparrow 0.2}$ | $\mathbf{82.3}_{\uparrow 0.9}$ | 80.7 | 78.9 | 81.4 | 78.9 | 80.3 | 80.4 |
| | MMVP | $75.6_{\uparrow 2.7}$ | $\mathbf{77.3}_{\uparrow 2.0}$ | 72.9 | 69.7 | 75.3 | 69.3 | 73.3 | 77.0 |
| | MMHal | $3.86_{\uparrow 0.06}$ | $\mathbf{3.88}_{\uparrow 0.05}$ | 3.80 | 3.79 | 3.83 | 3.81 | 3.72 | 3.76 |
| | RealWorldQA | $69.7_{\uparrow 0.9}$ | $69.2_{\uparrow 2.4}$ | 68.8 | 66.1 | 66.8 | 70.4 | 61.3 | 66.7 |
| | Avg. | $60.7_{\uparrow 2.1}$ | $\mathbf{61.4}_{\uparrow 1.2}$ | 58.6 | 57.4 | 60.2 | 58.3 | 57.3 | 59.7 |
| Image Reasoning | MathVista$_{testmini}$ | $74.0_{\uparrow 6.2}$ | $\mathbf{76.1}_{\uparrow 1.0}$ | 67.8 | 69.6 | 75.1 | 73.5 | 73.6 | 74.2 |
| | MathVision$_{testmini}$ | $30.6_{\uparrow 8.5}$ | $34.2_{\uparrow 1.3}$ | 22.1 | 24.3 | 32.9 | 30.7 | 27.9 | 36.2 |
| | MathVerse$_{testmini}$ | $49.7_{\uparrow 5.2}$ | $52.9_{\uparrow 0.8}$ | 44.5 | 47.2 | 52.1 | 51.9 | 50.2 | 54.7 |
| | MMMU$_{val}$ | $55.2_{\uparrow 4.6}$ | $\mathbf{58.8}_{\uparrow 3.3}$ | 50.6 | 53.8 | 55.5 | 50.5 | 52.7 | 57.4 |
| | EMMA$_{mini}$ | $28.3_{\uparrow 3.5}$ | $28.5_{\uparrow 1.3}$ | 24.8 | 26.0 | 29.8 | 34.5 | 31.0 | 28.0 |
| | Blind | $48.2_{\downarrow 1.1}$ | $\mathbf{51.3}_{\uparrow 2.1}$ | 49.3 | 48.7 | 49.2 | 38.0 | 48.1 | 47.5 |
| | V* | $81.5_{\uparrow 2.5}$ | $81.7_{\downarrow 1.6}$ | 79.0 | 77.4 | 83.3 | 78.0 | 61.8 | 68.6 |
| | VisuLogic | $28.0_{\uparrow 1.1}$ | $27.3_{\uparrow 1.8}$ | 26.9 | 26.0 | 25.5 | 31.0 | 26.3 | 25.4 |
| | ZeroBench$_{sub}$ | $16.5_{\uparrow 2.8}$ | $\mathbf{19.8}_{\uparrow 0.9}$ | 13.7 | 15.8 | 18.9 | 16.7 | 12.6 | 16.2 |
| | Avg. | $45.8_{\uparrow 3.7}$ | $\mathbf{47.8}_{\uparrow 0.9}$ | 42.1 | 43.2 | 46.9 | 46.2 | 42.7 | 45.3 |
| Chart Understanding | ChartQA | $\mathbf{85.4}_{\uparrow 1.4}$ | $84.2_{\downarrow 0.6}$ | 84.0 | 84.4 | 84.8 | 79.8 | 82.1 | 82.7 |
| | OCRBench | $86.1_{\uparrow 1.0}$ | $\mathbf{86.9}_{\uparrow 0.2}$ | 85.1 | 85.3 | 86.7 | 75.9 | 82.9 | 85.8 |
| | AI2D | $83.2_{\uparrow 0.6}$ | $\mathbf{83.8}_{\uparrow 0.2}$ | 82.6 | 79.9 | 83.6 | 80.4 | 82.3 | 82.9 |
| | Charxiv$_{reasoning}$ | $43.1_{\uparrow 1.7}$ | $\mathbf{47.3}_{\uparrow 1.5}$ | 41.4 | 41.9 | 45.8 | 36.1 | 40.2 | 43.4 |
| | Avg. | $74.5_{\uparrow 1.2}$ | $\mathbf{75.6}_{\uparrow 0.4}$ | 73.3 | 72.9 | 75.2 | 68.1 | 71.9 | 73.7 |
| Video Reasoning | MMVU$_{val}$ | $\mathbf{51.7}_{\uparrow 6.0}$ | $50.9_{\uparrow 1.1}$ | 45.7 | 49.4 | 49.8 | 50.7 | 50.8 | 51.2 |
| | VideoMMMU | $\mathbf{54.7}_{\uparrow 7.8}$ | $54.4_{\uparrow 1.2}$ | 46.9 | 49.1 | 53.2 | 52.6 | 52.6 | 51.9 |
| | Avg. | $\mathbf{53.2}_{\uparrow 6.9}$ | $52.7_{\uparrow 1.2}$ | 46.3 | 49.3 | 51.5 | 51.7 | 51.7 | 51.6 |
| VLM Agent | OSWorld | $\mathbf{3.5}_{\uparrow 0.8}$ | 0.0 | 2.7 | – | 0.0 | 0.0 | 0.0 | 0.0 |
| | Online-Mind2Web | $\mathbf{16.3}_{\uparrow 2.3}$ | $16.3_{\uparrow 1.0}$ | 14.0 | – | 15.3 | – | – | – |
| | Avg. | $\mathbf{9.9}_{\uparrow 1.5}$ | $8.2_{\uparrow 0.5}$ | 8.4 | – | 7.7 | – | – | – |
| Visual RewardBench | VLRewardBench | $\mathbf{61.5}_{\uparrow 10.8}$ | $59.8_{\uparrow 12.5}$ | 50.7 | 52.1 | 47.3 | 29.3 | 53.8 | 46.1 |
| | MM-RLHF | $\mathbf{74.7}_{\uparrow 2.9}$ | $70.0_{\uparrow 3.0}$ | 71.8 | 71.8 | 67.0 | 35.3 | 65.9 | 61.8 |
| | Avg. | $\mathbf{68.1}_{\uparrow 6.8}$ | $64.9_{\uparrow 7.7}$ | 61.3 | 62.0 | 57.2 | 32.3 | 59.8 | 53.9 |
| Overall Avg. | | $57.38_{\uparrow 3.1}$ | $\mathbf{58.23}_{\uparrow 1.47}$ | 54.28 | 54.37 | 56.76 | 52.67 | 54.16 | 53.09 |

reasoning tasks, matching or even surpassing specialized policy models. Together, these effects produce a **single model** that simultaneously strengthens both policy and critic capabilities within a single architecture.

# 3 MAIN RESULTS: CRITIC TRAINING AND TEST-TIME SCALING

## 3.1 CRITIC TRAINING

### 3.1.1 BASELINES

We use **Qwen-2.5-VL-7B** (Bai et al., 2025) as the baseline model. To evaluate the policy capability of critic training, we compare against four strong reasoning VLMs derived from in-domain policy training based on Qwen-2.5-VL-7B: **ThinkLite-VL-7B** (Wang et al., 2025e), **Vision-R1-7B** (Huang et al., 2025b), **MM-Eureka-7B** (Meng et al., 2025), and **VL-Rethinker-7B** (Wang et al., 2025a). Vision-R1 is first initialized by distilling the reasoning knowledge from Deepseek-R1, followed by RFT on 16K visual understanding and reasoning examples. The other three models are directly trained via RFT on top of Qwen-2.5-VL-7B, using varying amounts of diverse visual understanding and reasoning data, without relying on knowledge distillation.

For LLaVA-Critic-R1, we train two variants: (1) a base model version trained on top of Qwen-2.5-VL-7B using only 40k critic data to assess the policy improvement brought by critic training alone, and (2) a policy model version trained by further applying critic training on ThinkLite-VL-7B using same 40k critic data, thereby equipping the reasoning model with both policy and critic capabilities—referred to as LLaVA-Critic-R1+ from policy model in Table 3.

### 3.1.2 BENCHMARK EVALUATION RESULTS

We conduct a comprehensive evaluation of all VLMs across six dimensions, encompassing a total of 26 visual benchmarks: **(1) Perception & General VQA**: Blink (Fu et al., 2024),

HallusionBench$_{\text{Image}}$ (Guan et al., 2024), MMStar (Chen et al., 2024), MMBench (Liu et al., 2024a), MMVP (Tong et al., 2024), MMHal (Sun et al., 2023), and RealWorldQA; **(2) Image Reasoning**: MathVista (Lu et al., 2024), MathVision (Wang et al., 2024a), MathVerse (Zhang et al., 2024b), MMMU (Yue et al., 2024), EMMA (Hao et al., 2025), Blind (Rahmanzadehgervi et al., 2024), V$^*$ (Wu & Xie, 2023), VisuLogic (Xu et al., 2025), and ZeroBench (Roberts et al., 2025); **(3) Chart Understanding**: ChartQA (Masry et al., 2022), OCRBench (Liu et al., 2024b), AI2D (Kembhavi et al., 2016), and Charxiv reasoning (Wang et al., 2024d); **(4) Video Reasoning**: MMVU (Zhao et al., 2025b) and Video-MMMU (Hu et al., 2025); **(5) VLM Agent**: OSWorld (Xie et al., 2024a) and Online-Mind2Web (Xue et al., 2025); **(6) Visual RewardBench**: VLRewardBench (Li et al., 2025) and MM-RLHF (Zhang et al., 2025c).

During evaluation, all models follow a standardized thinking template format in Table 2 to ensure consistency and fair comparison. For Qwen-2.5-VL-7B, we report two variants: one performing direct answer generation without the template (denoted as Qwen-2.5-VL-7B-Non-Thinking), and the other using the thinking template for long-cot reasoning to enhance the performance during inference (denoted as Qwen-2.5-VL-7B-Thinking). The detailed evaluation results are presented in Table 3.

On Perception & General VQA tasks, critic training demonstrates a substantial improvement in the policy capabilities of VLMs. LLaVA-Critic-R1 and LLaVA-Critic-R1+ outperform their respective base models (Qwen-2.5-VL and ThinkLite-VL) with average improvements of +3.6% and +2.0%, winning 6/7 benchmarks. On complex image reasoning tasks, LLaVA-Critic-R1+ leads on 4/9 benchmarks (MathVista (76.1), MMMU (58.8), Blind (51.3), and ZeroBench (19.8)) and shows consistent gains, while LLaVA-Critic-R1 surpasses Qwen-2.5-VL on 8/9, averaging +7.25% improvement. In chart understanding, LLaVA-Critic-R1+ excels on OCRBench (86.9), AI2D (83.8), and Charxiv (47.3), and LLaVA-Critic-R1 perform best on ChartQA (85.4). We observe that critic training can also significant boost video reasoning ability. Evaluations on two video reasoning benchmarks show that the base version of LLaVA-Critic-R1 attains the best performance on both. Specifically, it improves upon Qwen-2.5-VL from 45.7 to 51.7 on MMVU and from 46.9 to 54.7 on Video-MMMU, substantially outperforming other reasoning model baselines.

Surprisingly, we find that critic training also enhances the model's capabilities as a Graphic User Interface (GUI) agent of Qwen-2.5-VL. On OSWorld, performance improves from 2.7 to 3.5, and on Online-Mind2Web, it increases from 14.0 to 16.3, demonstrating the improved grounding and decision-making capabilities in complex, long-horizon tasks. It's worth noting that we don't use any GUI images/data (screenshots/trajectories) that required typically by other agent models like UI-TARS (Qin et al., 2025). In contrast, reasoning models with policy training lose their ability to function effectively as agents—all VL reasoning models fail to complete tasks correctly on OSWorld. In such cases, applying critic training does not recover the agent capabilities of these models.

For critic ability, both versions of LLaVA-Critic-R1 demonstrate strong performance as evaluators, achieving average scores of 68.1 and 64.9, respectively. Notably, we observe that policy training significantly degrades the critic capabilities of VLMs, as evidenced by the consistently poor performance of all reasoning VLMs on the visual reward benchmarks. In contrast, critic training effectively enhances both the critic and policy capabilities of the model.

**More base model experiments**   We further validate the generality of critic RL training for enhancing policy capability by applying it to both Mimo-VL and LLaMA-3.2-Vision using the same dataset. The corresponding experimental results are reported in Table 6 and 7 of Appendix A and Appendix B, respectively. We observe that critic RL training consistently yields significant improvements in both policy and critic performance, even when initialized from different base models. Notably, LLaVA-Critic-R1+ trained on Mimo-VL-RL achieves 7B-level state-of-the-art performance across multiple benchmarks, including MMMU (71.9), MathVista (82.1), MathVerse (74.1), and CharXiv-Reasoning (62.5). These findings further substantiate the broad applicability and effectiveness of critic RL training in strengthening policy capability.

### 3.2 TEST-TIME SCALING WITH SELF-CRITIC

Given the strong dual capabilities of LLaVA-Critic-R1 as both a policy and a critic, this section further explores its potential as a reward model by employing it for test-time scaling. This strategy aims to enhance the policy performance of LLaVA-Critic-R1 through self-critic during inference.

Table 4: Performance of test-time scaling with self-critic on 5 visual reasoning and understanding benchmarks. LLaVA-Critic-R1 and LLaVA-Critic-R1+ achieve average performance improvements of 16.5% and 11.1%, respectively, substantially outperforming both the majority vote baseline and approaches that employ the base model as the critic.

| | MathVista | MathVerse | MathVision | MMMU | MMstar |
|---|---|---|---|---|---|
| LLaVA-Critic-R1 | 74.0 | 49.7 | 30.6 | 55.2 | 65.1 |
| + Best-of-128 (GT) | 94.8 | 87.5 | 97.7 | 95.9 | 93.6 |
| + Best-of-128 (Majority Vote) | 76.4 | 54.7 | 32.9 | 60.0 | 67.4 |
| + Best-of-128 (Qwen-Critic) | 77.1 | 55.2 | 34.7 | 61.9 | 67.9 |
| + Best-of-128 (Self-Critic) | 78.9 | 60.9 | 44.1 | 66.4 | 69.7 |
| LLaVA-Critic-R1+ | 76.1 | 52.9 | 32.4 | 58.8 | 65.7 |
| + Best-of-128 (GT) | 92.7 | 86.2 | 94.1 | 94.3 | 89.7 |
| + Best-of-128 (Majority Vote) | 77.3 | 56.9 | 35.2 | 61.6 | 67.2 |
| + Best-of-128 (ThinkLite-Critic) | 77.9 | 57.5 | 36.1 | 62.4 | 67.6 |
| + Best-of-128 (Self-Critic) | 78.7 | 61.3 | 41.7 | 65.9 | 70.1 |

**Implementation**  We begin by employing LLaVA-Critic-R1 as a policy model to perform $n$ rounds of thinking-based inference, generating $n$ candidate trajectories for each question. Subsequently, LLaVA-Critic-R1 serves as a critic model to select the best response among the $n$ candidates through a recursive pairwise comparison process. Specifically, all responses are paired, and for each pair, LLaVA-Critic-R1 evaluates and selects the more preferred one, resulting in $n/2$ candidates. This selection procedure is repeated iteratively until a single response remains, which is then designated as the final answer. Since LLaVA-Critic-R1 is trained on data containing tie cases, it may sometimes assign equal preference scores to both responses in a pair. In such cases, we randomly select one of the two responses to proceed to the next round. To ensure sufficient diversity among the generated trajectories during the thinking-based inference, we set the temperature parameter to 0.9.

**Baselines**  We introduce three critic methods as baselines. The first is the **Majority Vote**, where we extract the predicted answers from the $n$ generated responses and select the most frequently occurring answer as the final output. The second is the **Base Model Critic**, where we use the corresponding base models—Qwen-2.5-VL-7B for the base version and ThinkLite-VL-7B for the policy version—as critic models to perform pairwise comparisons over the $n$ responses, recursively selecting one final answer. Besides, we include a **Ground Truth** Oracle baseline to serve as an upper bound reference: if any of the $n$ generated answers match the ground-truth answer, the question is considered correctly answered.

**Results**  We run 128 inferences for both LLaVA-Critic-R1 and LLaVA-Critic-R1+, and evaluate the Best-of-128 performance under different critic strategies. We conduct experiment on five multimodel reasoning and understanding benchmarks: MathVista, MathVision, MathVerse, MMMU, and MMStar. The results are summarized in Table 4.

We observe that across all benchmarks, both versions of LLaVA-Critic-R1 achieve the highest performance when applying self-critic-based test-time scaling. Compared to the Majority Vote and Base Model Critic baselines, self-critic consistently yields substantial improvements. This highlights that, after dedicated critic training, LLaVA-Critic-R1 acquires significantly enhanced critic capabilities, enabling it to act as an effective critic without reliance on external supervision—thereby further improving policy performance through test-time self-critic.

Specifically, the base model version of LLaVA-Critic-R1 achieves the best performance on MathVista (78.9), MathVision (44.1), and MMMU (66.4), while the policy model version performs best on MathVerse (61.3) and MMStar (70.1). It is worth noting that although LLaVA-Critic-R1+ exhibits stronger policy capabilities and outperforms the base model across various reasoning benchmarks, its weaker critic ability limits its effectiveness during test-time self-critic scaling, resulting in no significant performance advantage over the base model version in this setting.

Furthermore, despite the clear improvements brought by self-critic scaling, there remains a substantial gap between its performance and the ground truth (GT) upper bound. In many cases, the model still fails to identify the correct answer from 128 generated responses, indicating that the current critic ability remains far from optimal. Importantly, such limitations are not captured by Visual RewardBench evaluations, underscoring the need for further advances in critic modeling.

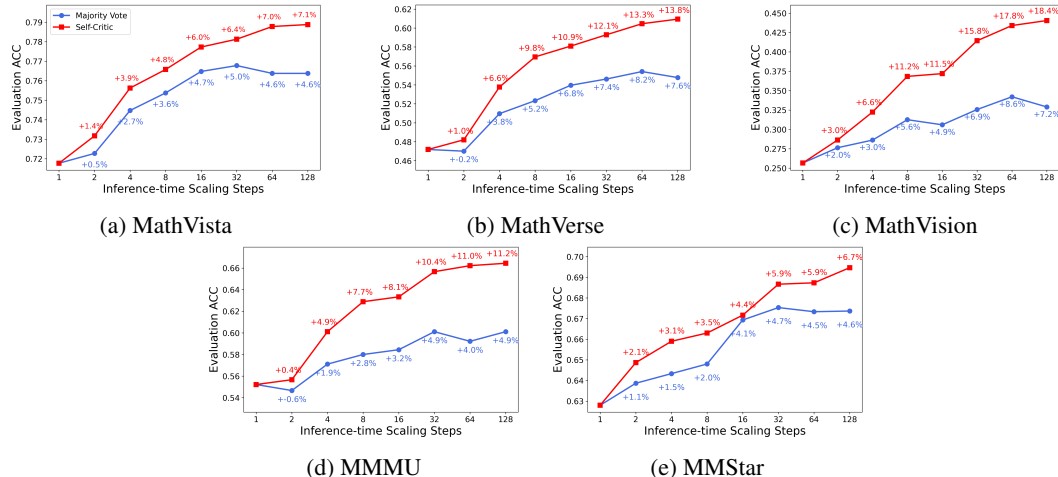

(a) MathVista      (b) MathVerse      (c) MathVision

(d) MMMU      (e) MMStar

Figure 2: Test-time scaling curve of self-critic and majority vote on 5 visual reasoning and understanding benchmarks with LLaVA-Critic-R1. Self-critic yields significantly greater performance gains than majority vote, and reaches performance plateau at a later sampling scale.

We also present the test-time scaling curve of the LLaVA-Critic-R1 base model across different benchmarks in Figure 2. The results demonstrate that as the scaling size increases from 2 to 64, the performance continues to improve, attributed to the model's self-critic capability. The performance saturates around a scaling size of 128. In contrast, the Majority Vote strategy reaches its performance ceiling much earlier, between scaling sizes 16 and 64, highlighting the advantage of self-critic during test-time scaling.

# 4 WHY CRITIC TRAINING CAN IMPROVE POLICY CAPABILITY?

In this section, we conduct experiments to investigate the reasons why critic RL training enhances the policy capability of VLMs. Furthermore, in Appendix C and D, we present systematic studies that (i) explore how to effectively train a model that achieves both strong policy and critic capability when given access to both critic and policy data, and (ii) examine the impact of SFT on critic model performance under the assumption of access to high-quality chain-of-thought (CoT) critic data. Besides, we also provide an analysis of the relationship between the model's critic capability and its policy performance during critic RL training process in Appendix E.

To elucidate the underlying reasons why RL-based critic training enhances a model's policy abilities, we consider two core hypotheses:

- **Enhanced Visual Perception**: Our constructed 40K critic training dataset consists of response pairs generated by various VLMs on natural image captioning and question-answering tasks. This setup requires the model to autonomously determine which response is more aligned with the image and to identify hallucinated content, thereby enhancing the model's visual perception capabilities.
- **Structured Reasoning**: The format reward in GRPO training encourages the model to strictly follow a "think-then-answer" generation pattern. This enforced reasoning structure strengthens the model's step-by-step reasoning ability, which is critical for improving policy effectiveness in complex tasks.

To validate our two hypotheses, we design and conduct two sets of controlled experiments. First, during the inference stage, we apply the thinking template from Table 2 to the Qwen-2.5-VL-7B, forcing it to perform a "think-then-answer" inference process without any additional training. Second, using 40K critic data for GRPO training, we set the weight of the format reward to 1 and the weight of the answer reward to 0. This allows the model to learn exclusively how to follow the "think-then-answer" format, without acquiring any critic-related knowledge. The experimental results are presented in Table 5. With the exception of the Qwen-2.5-VL-7B-non-thinking mode, all models are prompted with the thinking template during inference.

Table 5: Performance of Qwen-2.5-VL-7B base models, format reward only RL trained models (Format-RFT), and models with full critic training (LLaVA-Critic-R1). Bolded values indicate the best performance within each benchmark category.

| Benchmark | | Qwen (Non-Thinking) | Qwen (Thinking) | Format-RFT | LLaVA-Critic-R1 |
|---|---|---|---|---|---|
| Perception & General VQA | Blink | 54.2 | 53.9 | 54.2 | 58.3 |
| | HallusionBench$_{Image}$ | 67.9 | 67.2 | 66.9 | 71.3 |
| | MMStar | 61.7 | 62.5 | 62.9 | 65.1 |
| | MMBench | 80.7 | 78.9 | 77.9 | 80.9 |
| | MMVP | 72.9 | 69.7 | 70.3 | 75.6 |
| | MMHal | 3.80 | 3.79 | 3.72 | 3.86 |
| | RealWorldQA | 68.8 | 66.1 | 66.2 | 69.7 |
| | Avg. | 58.6 | 57.4 | 57.4 | **60.7** |
| Image Reasoning | MathVista$_{testmini}$ | 67.8 | 69.6 | 70.2 | 74.0 |
| | MathVision$_{testmini}$ | 22.1 | 24.3 | 25.6 | 30.6 |
| | MathVerse$_{testmini}$ | 44.5 | 47.2 | 47.9 | 49.7 |
| | MMMU$_{val}$ | 50.6 | 53.8 | 54.7 | 55.2 |
| | EMMA$_{mini}$ | 24.8 | 26.0 | 26.8 | 28.3 |
| | Blind | 49.3 | 48.7 | 48.9 | 48.2 |
| | V$^*$ | 79.0 | 77.4 | 79.6 | 81.5 |
| | VisuLogic | 26.9 | 26.0 | 25.0 | 28.0 |
| | ZeroBench$_{sub}$ | 13.7 | 15.8 | 16.5 | 16.5 |
| | Avg. | 42.1 | 43.2 | 43.9 | **45.8** |
| Chart Understanding | ChartQA | 84.0 | 84.4 | 84.2 | 85.4 |
| | OCRBench | 85.1 | 85.3 | 85.5 | 86.1 |
| | AI2D | 82.6 | 79.9 | 79.8 | 83.2 |
| | Charxiv$_{reasoning}$ | 41.4 | 41.9 | 42.3 | 43.1 |
| | Avg. | 73.3 | 72.9 | 73.0 | **74.5** |
| Video Reasoning | MMVU$_{val}$ | 45.7 | 49.4 | 48.7 | 51.7 |
| | VideoMMMU | 46.9 | 49.1 | 50.3 | 54.7 |
| | Avg. | 46.3 | 49.3 | 49.5 | **53.2** |
| Visual RewardBench | VLRewardBench | 50.7 | 52.1 | 55.6 | 61.5 |
| | MM-RLHF | 71.8 | 71.8 | 69.6 | 74.7 |
| | Avg. | 61.3 | 62.0 | 62.6 | **68.1** |
| Overall Avg. | | 54.28 | 54.37 | 54.72 | **57.38** |

We find that in the setting with inference-time thinking alone, Qwen-2.5-VL-7B-Thinking shows a significant performance improvement in both visual reasoning (42.1 to 43.2) and video reasoning (46.3 to 49.3) domains. After further GRPO training with the format reward to enforce stricter adherence to the thinking template, the model's performance in these two domains improves further to 43.9 and 49.5, respectively. However, on other tasks that require stronger visual perception capabilities, the model's performance degrades under both of these settings.

In contrast, following critic training, LLaVA-Critic-R1 demonstrates a notable performance increase on several Perception & General VQA and Chart Understanding benchmarks, indicating that its perception capabilities are enhanced. Furthermore, because critic training also implicitly forces the model to follow the thinking template, LLaVA-Critic-R1's performance in the two reasoning domains is further boosted to 45.8 and 53.2. This suggests that enhancing perception capabilities also contributes to performance improvements on visual reasoning tasks.

These two experiments collectively validate our central hypothesis: the enhancement of policy ability through critic training stems from two synergistic factors: **a stricter ability to follow the thinking template** and **a stronger visual perception capability gained from the data**. These two abilities mutually reinforce each other, leading to a significant improvement in the model's overall policy performance.

## 5 RELATED WORK

**Multimodal Critic Model Training.** Critic model training has evolved from scalar approaches using Bradley-Terry modeling (Ouyang et al., 2022; Gao et al., 2023) to more sophisticated architectures. Critic model provides step-wise supervision for complex reasoning (Lightman et al., 2023; Cobbe et al., 2021), while generative critic models enable richer feedback through natural language (Mahan

et al., 2024; Zhang et al., 2024a; Zheng et al., 2023). For vision-language models, various approaches adopt distinct strategies for critic training. Wang et al. (2024c) and Wang et al. (2025c) integrate an external critic head into the VLM, which is trained via SFT to directly output step-wise scores during inference time. Zhang et al. (2025c) and Xiong et al. (2024) leverage knowledge distillation to train a generative critic model, thereby providing richer visual feedback for both pairwise and pointwise data. In this paper, we investigate an approach that restructures critic data into verifiable RL task and trains a generative critic model via RL, without relying on knowledge distillation. Zhang et al. (2025b) also explore this direction, but their method first applies knowledge distillation before RL training, and their analysis is limited to the impact on critic capability. In contrast, our work focuses on examining the effect of critic training on downstream general task performance, i.e. policy capability. Furthermore, as detailed in Section C, our ablation study offers a more systematic and comprehensive analysis of the integration between supervised fine-tuning (SFT) and RL.

**VLM Reasoning.** Reinforcement fine-tuning has emerged as a powerful paradigm for enhancing reasoning capabilities in vision-language models, extending successful approaches from text-only models like DeepSeek-R1 (DeepSeek-AI, 2025). Recent studies have demonstrated that reinforcement fine-tuning can facilitate advanced visual reasoning through various strategies, ranging from distilling reasoning knowledge from stronger models before applying RL (Huang et al., 2025a; Deng et al., 2025; Wan et al., 2025; Wei et al., 2025), to exploring multimodal training data selection methods (Meng et al., 2025; Peng et al., 2025; Wang et al., 2025e;a;d) and refining GRPO for multimodal RL (Wang et al., 2025a; Liu et al., 2025; Wan et al., 2025; Yao et al., 2025; Yuan et al., 2025). However, these works predominantly employ multimodal reasoning and understanding data with ground truth answers directly for reinforcement fine-tuning. In this paper, we explore RL training from the perspective of leveraging critic data, showing that critic reinforcement fine-tuning can also effectively boost a model's policy capability. Moreover, we provide empirical evidence that critic training and policy training are orthogonal in the context of reinforcement fine-tuning to further improve the policy ability of VLMs.

**Test-Time Scaling.** Recent advances demonstrate that scaling compute at inference time can be as effective as scaling model parameters (Snell et al., 2024), opening new avenues for improving model performance without retraining. Test-time scaling approaches broadly encompass: (1) search and sampling methods, where Monte Carlo Tree Search (Xie et al., 2024b; Gao et al., 2024) and compute-optimal strategies (Snell et al., 2024; Wu et al., 2024) surpass simple best-of-N baselines through adaptive compute allocation; (2) long-horizon chain-of-thought reasoning (Wei et al., 2022), exemplified by OpenAI o1 (OpenAI, 2024; Jaech et al., 2024) and DeepSeek-R1 (DeepSeek-AI, 2025), which generate extensive internal reasoning chains with performance scaling smoothly with thinking time; and (3) critic model-guided aggregation and verification (Lightman et al., 2024; Brown et al., 2024), where process reward models (Zhao et al., 2025a; Khalifa et al., 2025) and Q-learning-based verifiers (Qi et al., 2024) critic model (Zhao et al., 2025a; Khalifa et al., 2025; Wang et al., 2024c; 2025c) guide selection among multiple outputs. Our work bridges these paradigms by training a unified model with both strong policy and critic ability through reinforcement learning that enables effective self-critic at test time. Unlike existing approaches that rely on external verifiers or massive search, our method develops intrinsic judgment capabilities within the model itself, achieving test-time scaling through self-evaluation that surpasses traditional majority voting and approaches oracle performance on challenging visual reasoning tasks, offering a scalable alternative to compute-intensive search methods while contributing to the broader goal of inference-time co-scaling between policy models and critic models.

## 6 CONCLUSION

We find that critic RL training, originally intended to enhance the critic ability, can surprisingly deliver substantial improvement in policy performance, producing a single model that excel in both critic and policy roles. This is achieved by reorganizing preference-labeled critic data into verifiable training signals and applying RL training to directly optimize multimodal judgment. Results on a series of visual benchmarks further demonstrate the strong policy and critic abilities of the resulting models LLaVA-Critic-R1 and LLaVA-Critic-R1+. Furthermore, the strengthened critic enables test-time scaling for additional gains, highlighting its broader utility in multimodal learning. We hope the insights inspire further research to develop self-improvement AI, where the model can make useful judgment its own response, and thus provide feedback for self-improvement.

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

## A  CRITIC RL TRAINING ON MIMO-VL

In this section, we perform critic RL training based on the state-of-the-art reasoning vision-language model, MiMo-VL (Team et al., 2025a). Specifically, we utilize two versions of MiMo-VL—MiMo-VL-7B-SFT-2508 and MiMo-VL-7B-RL-2508—as base models to develop two variants of LLaVA-Critic-R1+. The evaluation results on 24 visual reasoning and understanding benchmarks are presented in Table 6.

Table 6: Compare the evaluation results of LLaVA-Critic-R1+ obtained via critic RL training from two versions of MiMo-VL across 24 visual reasoning and understanding benchmarks. The variant trained from MiMo-VL-7B-RL-2508 achieves new 7B-level state-of-the-art performance on MMMU (71.9), MathVista (82.1), MathVerse (74.1), and ChartXiv Reasoning (62.5).

| Benchmark | | MiMo-VL-SFT-7B | | MiMo-VL-RL-7B | |
|---|---|---|---|---|---|
| | | LLaVA-Critic-R1 | Base | LLaVA-Critic-R1+ | Base |
| Perception & General VQA | Blink | $62.4_{\uparrow 1.9}$ | 60.5 | $63.5_{\uparrow 0.6}$ | 62.9 |
| | HallusionBench$_{Image}$ | $76.3_{\uparrow 1.3}$ | 75.0 | $76.9_{\downarrow 0.7}$ | 77.6 |
| | MMStar | $71.9_{\uparrow 0.4}$ | 71.5 | $74.2_{\uparrow 0.9}$ | 73.3 |
| | MMBench | $84.7_{\downarrow 0.2}$ | 84.9 | $85.5_{\uparrow 0.6}$ | 84.9 |
| | MMVP | $79.0_{\uparrow 0.4}$ | 78.6 | $81.1_{\downarrow 0.5}$ | 81.6 |
| | MMHal | $4.41_{\uparrow 0.18}$ | 4.23 | $4.42_{\uparrow 0.07}$ | 4.35 |
| | RealWorldQA | $69.8_{\downarrow 0.6}$ | 70.4 | $70.9_{\downarrow 0.4}$ | 71.3 |
| | Avg. | $64.1_{\uparrow 0.5}$ | 63.6 | $65.2_{\uparrow 0.1}$ | 65.1 |
| Image Reasoning | MathVista$_{testmini}$ | $81.9_{\uparrow 0.8}$ | 81.1 | $82.1_{\uparrow 0.5}$ | 81.6 |
| | MathVision$_{testmini}$ | $55.2_{\uparrow 1.6}$ | 53.6 | $63.5_{\uparrow 4.0}$ | 59.5 |
| | MathVerse$_{testmini}$ | $70.5_{\uparrow 2.4}$ | 68.1 | $74.1_{\uparrow 2.9}$ | 71.2 |
| | MMMU$_{val}$ | $69.7_{\uparrow 1.3}$ | 68.4 | $71.9_{\uparrow 1.6}$ | 70.3 |
| | EMMA$_{mini}$ | $36.8_{\downarrow 2.0}$ | 38.8 | $39.8_{\downarrow 0.5}$ | 40.3 |
| | Blind | $74.6_{\uparrow 0.1}$ | 74.5 | $76.9_{\uparrow 0.1}$ | 76.8 |
| | V$^*$ | $83.2_{\uparrow 1.1}$ | 82.1 | $85.3_{\uparrow 1.6}$ | 83.7 |
| | VisuLogic | $25.6_{\uparrow 0.4}$ | 25.2 | $26.7_{\uparrow 2.4}$ | 24.3 |
| | ZeroBench$_{sub}$ | $21.0_{\uparrow 2.2}$ | 18.8 | $21.3_{\uparrow 0.4}$ | 20.9 |
| | Avg. | $57.6_{\uparrow 0.9}$ | 56.7 | $60.2_{\uparrow 1.5}$ | 58.7 |
| Chart Understanding | ChartQA | $88.5_{\uparrow 0.6}$ | 87.9 | $90.1_{\uparrow 1.6}$ | 88.5 |
| | OCRBench | $85.0_{\uparrow 3.6}$ | 81.4 | $86.6_{\uparrow 3.8}$ | 82.8 |
| | AI2D | $85.4_{\downarrow 0.3}$ | 85.7 | $85.9_{\uparrow 0.1}$ | 85.8 |
| | Charxiv$_{reasoning}$ | $57.6_{\uparrow 2.8}$ | 54.8 | $62.5_{\uparrow 3.2}$ | 59.3 |
| | Avg. | $79.1_{\uparrow 1.6}$ | 77.5 | $81.3_{\uparrow 2.1}$ | 79.1 |
| Video Reasoning | MMVU$_{val}$ | $59.8_{\uparrow 0.6}$ | 59.2 | $61.2_{\uparrow 0.6}$ | 60.6 |
| | VideoMMMU | $55.1_{\uparrow 0.9}$ | 54.2 | $53.5_{\uparrow 1.4}$ | 52.1 |
| | Avg. | $57.5_{\uparrow 0.8}$ | 56.7 | $57.4_{\uparrow 1.0}$ | 56.4 |
| Visual RewardBench | VLRewardBench | $67.4_{\uparrow 5.6}$ | 61.8 | $66.4_{\uparrow 5.1}$ | 61.3 |
| | MM-RLHF | $77.6_{\uparrow 7.1}$ | 70.5 | $78.8_{\uparrow 10.6}$ | 68.2 |
| | Avg. | $72.5_{\uparrow 6.3}$ | 66.2 | $72.6_{\uparrow 7.8}$ | 64.8 |
| Overall Avg. | | $64.31_{\uparrow 1.34}$ | 62.97 | $65.96_{\uparrow 1.66}$ | 64.30 |

The experimental results demonstrate that both versions of LLaVA-Critic-R1+ exhibit substantial performance improvements over their respective base models across most of visual reasoning and understanding tasks. Notably, the variant based on MiMo-VL-7B-RL-2508 achieves SOTA performance among all 7B-scale VLMs on several challenging benchmarks, including MMMU (71.8), MathVista (82.1), MathVerse (74.1), and Charxiv Reasoning (62.5), surpassing previously reported best results by a significant margin.

Furthermore, on both VLM reward benchmarks, critic RL training yields consistently large gains, highlighting its effectiveness in enhancing the model's critic capability. In terms of average performance, LLaVA-Critic-R1+ improves upon its base models by 2.1% (MiMo-VL-7B-SFT-2508) and 2.6% (MiMo-VL-7B-RL-2508), respectively, establishing it as the most capable 7B-level vision-language model for reasoning to date. These results further substantiate the effectiveness of our approach in enhancing the policy capabilities of VLMs.

# B    CRITIC RL TRAINING ON LLaMA-3.2-VISION

In this section, we conduct critic RL training on Llama-3.2-11B-Vision-Instruct to further verify the effectiveness of our approach. Since Llama-3.2-11B-Vision-Instruct does not accept video inputs, we evaluate the policy capability of LLaVA-Critic-R1 on 20 visual understanding and reasoning benchmarks. As shown in Table 7, critic RL training consistently improves the policy performance of LLaMA-3.2-Vision, yielding an average gain of 2.6% across the 20 tasks. The improvements are particularly pronounced on reasoning tasks, with MMMU increasing by 13.4% (from 43.2 to 49.0) and MathVision by 30.9% (from 23.6 to 30.9).

Table 7: Compare the evaluation results of LLaVA-Critic-R1 obtained via critic RL training from Llama-3.2-11B-Vision-Instruct across 20 visual reasoning and understanding benchmarks.

|  | | Llama-3.2-11B-Vision-Instruct | |
| --- | --- | --- | --- |
| Benchmark | | LLaVA-Critic-R1 | Base |
| Perception & General VQA | Blink | $44.9_{\uparrow 1.6}$ | 43.3 |
| | HallusionBench$_{Image}$ | $63.1_{\uparrow 0.3}$ | 62.8 |
| | MMStar | $54.9_{\downarrow 1.3}$ | 56.2 |
| | MMBench | $77.6_{\uparrow 0.3}$ | 77.3 |
| | MMVP | $73.0_{\uparrow 1.2}$ | 71.9 |
| | MMHal | $3.12_{\uparrow 0.02}$ | 3.10 |
| | RealWorldQA | $62.1_{\uparrow 1.6}$ | 60.5 |
| | Avg. | $54.1_{\uparrow 0.5}$ | 53.6 |
| Image Reasoning | MathVista$_{testmini}$ | $48.9_{\uparrow 0.6}$ | 48.3 |
| | MathVision$_{testmini}$ | $30.9_{\uparrow 7.3}$ | 23.6 |
| | MathVerse$_{testmini}$ | $30.8_{\uparrow 1.3}$ | 29.5 |
| | MMMU$_{val}$ | $49.0_{\uparrow 5.8}$ | 43.2 |
| | EMMA$_{mini}$ | $24.8_{\uparrow 2.0}$ | 22.8 |
| | Blind | $32.0_{\uparrow 0.7}$ | 31.3 |
| | V$^*$ | $61.8_{\downarrow 0.5}$ | 62.3 |
| | VisuLogic | $26.7_{\uparrow 0.5}$ | 26.2 |
| | ZeroBench$_{sub}$ | $16.5_{\uparrow 0.4}$ | 16.1 |
| | Avg. | $35.7_{\uparrow 2.0}$ | 33.7 |
| Chart Understanding | ChartQA | $76.9_{\uparrow 0.3}$ | 76.6 |
| | OCRBench | $77.0_{\downarrow 0.5}$ | 77.5 |
| | AI2D | $78.5_{\uparrow 0.2}$ | 78.3 |
| | Charxiv$_{reasoning}$ | $35.3_{\uparrow 2.8}$ | 32.5 |
| | Avg. | $66.9_{\uparrow 0.7}$ | 66.2 |
| Overall Avg. | | $48.39_{\uparrow 1.22}$ | 47.17 |

# C    CRITIC/POLICY TRAINING SYNERGY

In this section, we conduct an ablation study to systematically explore how to best train a model that achieves both strong policy performance and enhanced critic capability, given access to both critic and policy data.

All our training is based on the Qwen-2.5-VL-7B model, using GRPO as the objective for RFT. For data sources, we use 40K pairwise critic data that we curated and 70K policy data from the ThinkLite-70K dataset, which includes multimodal math reasoning, natural image understanding, and chart understanding tasks. We explore three data configurations to assess their impact on model performance: **(1)** training with critic data only, **(2)** training with policy data only, and **(3)** joint training with both critic and policy data. In the first setting, the model trained solely on critic data

is referred to as LLaVA-Critic-R1. In the second, the model trained on ThinkLite-70K is denoted as ThinkLite-VL-7B. For the third setting involving both data types, we investigate three training strategies:

- Mixed Training: directly combining critic and policy data for RFT;
- Critic-then-Policy Training: starting from the LLaVA-Critic-R1, followed by RFT on policy data (ThinkLite-70K);
- Policy-then-Critic Training: initializing from ThinkLite-VL-7B and further training on 40K critic data.

In Table 8, we report the evaluation results of models trained with five different strategies across various benchmark categories. We find that LLaVA-Critic-R1, trained from the base model using only critic data, demonstrates the strongest critic capability, achieving the highest performance on visual reward benchmarks (68.1), along with a moderate improvement in policy performance. In contrast, ThinkLite-VL, trained solely on policy data, exhibits strong policy performance across multiple benchmarks (average score of 56.72), but its critic capability remains suboptimal (57.2) due to the absence of critic training.

Among the three models trained with both critic and policy data, we observe the following:

- Mixed training improves both policy and critic capabilities to some extent, but falls short of the performance achieved by models trained on individual datasets alone. Notably, its policy performance is significantly lower than that of ThinkLite-VL-7B.
- Critic-then-policy training slightly enhances policy performance (from 56.41 to 56.53) but causes a notable degradation in critic ability (from 68.1 to 62.9).
- **Policy-then-critic training emerges as the optimal strategy**: continuing critic training from the ThinkLite-VL checkpoint not only improves policy performance (from 56.72 to 57.54) but also substantially recovers the critic capability that is weakened by policy training (from 57.2 to 64.9), thereby achieving a better balance between the two competencies.

## D   ON THE BEST UTILIZATION OF CRITIC DATA

Given that supervised fine-tuning (SFT) has traditionally been the dominant paradigm for training critic models, in this section, we conduct an ablation study to investigate the impact of SFT on critic model performance under the assumption of access to high-quality chain-of-thought (CoT) critic data.

We first explore three different SFT strategies: (1) fine-tuning only the LLM parameters, (2) jointly fine-tuning the LLM and the projection layer, and (3) full-parameter fine-tuning. Based on the checkpoints obtained from each strategy, we subsequently perform critic RFT. For SFT, we use the LLaVA-Critic-113K dataset (Xiong et al., 2025), which comprises 73K pointwise critic data and 40K pairwise critic data. Each data point includes not only the ground truth score but also a reasoning explanation generated by GPT to justify the assigned score. For the RFT stage, we consistently use 40K pairwise critic samples with ground truth labels only. To ensure a fair comparison, we also performe an additional SFT experiment using only the 40K pairwise critic data from LLaVA-Critic-113K, matching the data size used in LLaVA-Critic-R1.

We still conduct evaluations on a range of visual reasoning and understanding benchmarks, with results presented in Table 9. The results indicate that applying SFT-based knowledge distillation does not lead to significant improvements in model performance on the two visual reward benchmarks (VL-RewardBench and MM-RLHF). Moreover, this form of distillation substantially degrades the model's policy capability, resulting in poor performance across several general visual benchmarks. **This observation suggests that relying solely on SFT may impair the model's generalization ability—a conclusion that has also been reported in recent study (Zhang et al., 2025a)**.

In contrast, models trained via RFT based on the three different SFT checkpoints show substantial improvements on the reward benchmarks. Notably, the model initialized from the LLM-only SFT checkpoint achieves the highest performance on MM-RLHF (77.0). However, on VL-RewardBench, LLaVA-Critic-R1 —which is trained directly via RFT without any prior SFT—achieves the best

Table 8: Ablation study of training strategies using critic and policy data. Bolded values indicate the best performance within each benchmark category.

| Benchmark | | Critic Training | Policy Training | Critic Training + Policy Training | | |
|---|---|---|---|---|---|---|
| | | LLaVA-Critic-R1 | ThinkLite-VL-7B | Mixed Data | Critic –> Policy | Policy –> Critic |
| General visual question answering | Blink | 58.3 | 58.5 | 56.9 | 56.4 | 59.0 |
| | HallusionBench$_{Image}$ | 71.3 | 70.9 | 69.9 | 71.0 | 72.3 |
| | MMStar | 65.1 | 65.0 | 63.8 | 64.1 | 65.7 |
| | MMBench | 80.9 | 81.4 | 80.3 | 81.4 | 82.3 |
| | MMVP | 75.6 | 75.3 | 74.0 | 70.0 | 77.3 |
| | MMHal | 3.86 | 3.83 | 3.81 | 3.82 | 3.88 |
| | RealWorldQA | 69.7 | 66.8 | 69.0 | 69.9 | 69.2 |
| | Avg. | 60.7 | 60.2 | 59.7 | 59.5 | **61.4** |
| Image Reasoning | MathVista$_{testmini}$ | 74.0 | 75.1 | 75.0 | 74.9 | 76.1 |
| | MathVision$_{testmini}$ | 30.6 | 32.9 | 28.3 | 30.9 | 32.4 |
| | MathVerse$_{testmini}$ | 49.7 | 52.1 | 49.8 | 51.4 | 52.9 |
| | MMMU$_{val}$ | 55.2 | 55.5 | 55.8 | 57.7 | 58.8 |
| | EMMA$_{mini}$ | 28.3 | 29.8 | 26.0 | 27.0 | 28.5 |
| | Blind | 48.2 | 49.2 | 49.5 | 52.1 | 51.3 |
| | V$^*$ | 81.5 | 83.3 | 80.6 | 80.9 | 81.7 |
| | VisuLogic | 28.0 | 25.5 | 25.4 | 26.3 | 27.3 |
| | ZeroBench$_{sub}$ | 16.5 | 18.9 | 15.3 | 19.2 | 19.8 |
| | Avg. | 45.8 | 46.9 | 45.1 | 46.7 | **47.7** |
| Chart Understanding | ChartQA | 85.4 | 84.8 | 84.1 | 84.9 | 84.2 |
| | OCRBench | 86.1 | 86.7 | 86.3 | 86.8 | 86.9 |
| | AI2D | 83.2 | 83.6 | 82.5 | 83.4 | 83.8 |
| | Charxiv$_{reasoning}$ | 43.1 | 45.8 | 44.2 | 46.5 | 47.3 |
| | Avg. | 74.5 | 75.2 | 74.3 | 75.4 | **75.6** |
| Video Reasoning | MMVU$_{val}$ | 51.7 | 49.8 | 49.1 | 50.1 | 50.9 |
| | VideoMMMU | 54.7 | 53.2 | 52.1 | 55.0 | 54.4 |
| | Avg. | **53.2** | 51.5 | 50.6 | 52.6 | 52.7 |
| Visual RewardBench | VLRewardBench | 61.5 | 47.3 | 58.6 | 58.1 | 59.8 |
| | MM-RLHF | 74.7 | 67.0 | 71.2 | 67.6 | 70.0 |
| | Avg. | **68.1** | 57.2 | 64.9 | 62.9 | 64.9 |
| Overall Avg. w. RewardBench | | 57.38 | 56.76 | 56.31 | 57.06 | **58.16** |
| Overall Avg. w.o. RewardBench | | 56.41 | 56.72 | 55.53 | 56.53 | **57.54** |

result. Besides, LLaVA-Critic-R1 consistently maintains the strongest policy performance among all evaluated models on general visual tasks.

These results suggest that if the primary goal is to build a critic model with strong judgment capabilities, performing long-CoT SFT followed by RFT offers certain advantages over cold-start RFT. However, **if the objective is to achieve a balanced capability of both policy and critic, the most effective strategy is to start from a strong policy/reasoning model and directly apply cold-start RFT.**

# E   ON THE CORRECTION BETWEEN CRITIC AND POLICY ABILITY

In this section, we focus on analyzing the relationship between the model's critic capability and its policy performance during critic training. Figure 3 illustrates the evolution of critic and policy performance over the critic RFT steps. Here, critic performance is measured on two visual reward benchmark, while policy performance is gauged on other general visual tasks in Table 3.

A striking observation is the strong positive correlation between the two curves during the initial phase of training (approximately steps 0 to 200). The continuous improvement in critic performance (red line) is accompanied by a steady increase in policy performance (blue line). In the latter half of the training (approximately steps 200 to 400), this strong correlation appears to weaken. The critic performance curve begins to show fluctuations, while the policy performance curve peaks around step 350 and then starts to decline. This divergence could be attributed to various factors, such as the onset of overfitting, where LLaVA-Critic-R1, despite performing well on the reward benchmark, may provide feedback signals that are no longer optimal for the general task, consequently leading to a degradation in policy performance. Nevertheless, the overall trend throughout the training process remains largely consistent, confirming a significant relationship between critic and policy capabilities.

Table 9: Ablation study of critic data utilization strategies. Models are trained with SFT, RFT, or both. Bolded values indicate the best performance within each benchmark category.

| | Benchmark | Base Qwen-2.5-VL-7B | SFT LLM | SFT LLM+MMP | SFT Full | SFT Pairwise-Only | SFT+RFT LLM | SFT+RFT LLM+MMP | SFT+RFT Full | RFT LLaVA-Critic-R1 |
|---|---|---|---|---|---|---|---|---|---|---|
| General visual question answering | Blink | 54.2 | 53.0 | 52.9 | 54.5 | 51.4 | 56.4 | 56.5 | 55.0 | 58.3 |
| | HallusionBench$_{Image}$ | 67.9 | 68.8 | 68.2 | 67.5 | 66.9 | 70.9 | 70.1 | 68.0 | 71.3 |
| | MMStar | 61.7 | 61.1 | 61.1 | 61.7 | 59.7 | 61.6 | 63.0 | 60.7 | 65.1 |
| | MMBench | 80.7 | 76.6 | 77.1 | 77.1 | 77.5 | 80.5 | 80.1 | 79.8 | 80.9 |
| | MMVP | 72.9 | 69.3 | 68.0 | 70.7 | 69.0 | 74.3 | 72.7 | 72.3 | 75.6 |
| | MMHal | 3.80 | 3.82 | 3.80 | 3.78 | 3.79 | 3.84 | 3.83 | 3.81 | 3.86 |
| | RealWorldQA | 68.8 | 64.8 | 65.8 | 64.6 | 63.7 | 67.0 | 68.6 | 69.5 | 69.7 |
| | Avg. | 58.6 | 56.8 | 56.7 | 57.1 | 56.0 | 59.2 | 59.3 | 58.4 | **60.7** |
| Image Reasoning | MathVista$_{testmini}$ | 67.8 | 66.4 | 65.3 | 67.3 | 66.5 | 72.9 | 70.3 | 69.8 | 74.0 |
| | MathVision$_{testmini}$ | 22.1 | 31.3 | 31.3 | 28.9 | 29.3 | 27.6 | 31.6 | 29.6 | 30.6 |
| | MathVerse$_{testmini}$ | 44.5 | 48.6 | 47.8 | 47.8 | 45.9 | 49.3 | 48.8 | 47.1 | 49.7 |
| | MMMU$_{val}$ | 50.6 | 53.1 | 53.1 | 54.2 | 53.7 | 52.9 | 56.4 | 56.0 | 55.2 |
| | EMMA$_{mini}$ | 24.8 | 25.5 | 23.8 | 25.8 | 26.0 | 26.3 | 26.8 | 28.3 | 28.3 |
| | Blind | 49.3 | 49.5 | 49.4 | 48.5 | 46.7 | 49.8 | 49.8 | 48.8 | 48.2 |
| | V$^{*}$ | 79.0 | 80.6 | 78.5 | 79.6 | 78.0 | 79.1 | 79.0 | 78.0 | 81.5 |
| | VisuLogic | 26.9 | 25.3 | 25.7 | 25.2 | 24.8 | 24.5 | 24.8 | 26.3 | 28.0 |
| | ZeroBench$_{sub}$ | 13.7 | 16.2 | 14.1 | 15.9 | 15.3 | 15.3 | 16.5 | 16.8 | 16.5 |
| | Avg. | 42.1 | 44.1 | 43.2 | 43.7 | 42.9 | 44.2 | 44.9 | 44.5 | **45.8** |
| Chart Understanding | ChartQA | 84.0 | 83.3 | 83.3 | 82.8 | 82.3 | 84.9 | 83.2 | 81.3 | 85.4 |
| | OCRBench | 85.1 | 85.1 | 85.8 | 84.8 | 85.5 | 85.9 | 86.3 | 86.9 | 86.1 |
| | AI2D | 82.6 | 77.9 | 78.0 | 78.1 | 79.7 | 81.9 | 81.2 | 81.5 | 83.2 |
| | Charxiv$_{reasoning}$ | 41.4 | 40.5 | 39.0 | 38.4 | 39.1 | 41.1 | 41.5 | 38.7 | 43.1 |
| | Avg. | 73.3 | 71.7 | 71.5 | 71.1 | 71.7 | 73.5 | 73.1 | 72.1 | **74.5** |
| Video Reasoning | MMVU$_{val}$ | 45.7 | 48.6 | 48.7 | 47.4 | 47.0 | 50.8 | 49.4 | 49.5 | 51.7 |
| | VideoMMMU | 46.9 | 50.1 | 49.9 | 50.0 | 49.7 | 52.5 | 53.0 | 52.9 | 54.7 |
| | Avg. | 46.3 | 49.4 | 49.3 | 48.7 | 48.4 | 51.7 | 51.2 | 51.2 | **53.2** |
| Visual RewardBench | VLRewardBench | 50.7 | 52.5 | 51.6 | 52.2 | 51.9 | 57.9 | 56.8 | 57.7 | 61.5 |
| | MM-RLHF | 71.8 | 72.5 | 72.9 | 71.5 | 72.1 | 77.0 | 75.3 | 71.2 | 74.7 |
| | Avg. | 61.3 | 62.5 | 62.3 | 61.9 | 62.0 | 67.5 | 66.1 | 64.5 | **68.1** |
| | Overall Avg. | 54.28 | 54.35 | 53.96 | 54.11 | 53.56 | 56.01 | 56.06 | 55.39 | **57.38** |

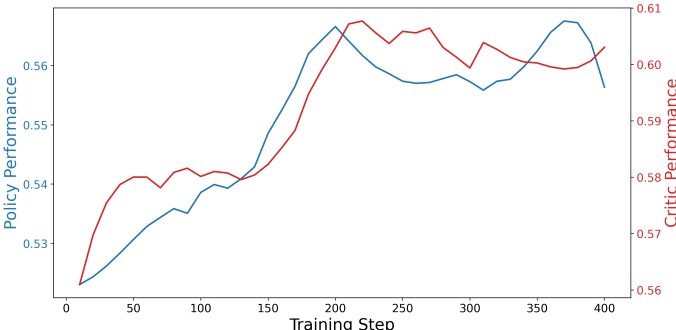

Figure 3: Correlation between LLaVA-Critic-R1's critic and policy performance. This figure plots the performance of LLaVA-Critic-R1's two key capabilities as a function of critic training steps. The blue curve represents the policy performance on general visual tasks, while the red curve denotes the critic performance on visual reward benchmarks. The strong positive correlation between the two performance metrics throughout the training process provides empirical evidence that improvements in the model's critic ability are directly linked to enhancements in its policy ability.

## F  CASE STUDY

In this section, we present case studies (Table 10 to 12) to further compare the reasoning behaviors of LLaVA-Critic-R1+ and the base model.

## G  THE USE OF LARGE LANGUAGE MODELS (LLMS)

In this work, large language models (LLMs) are used only to polish the writing and improve the clarity of language. Beyond this purpose, LLMs do not contribute to the method and experimental design, data analysis, or interpretation of results.

Table 10: An example where Qwen-2.5-VL follows the thinking template but fails in reasoning, while LLaVA-Critic-R1+ correctly interprets chart details and derives the accurate answer.

| Example 1 | |
|---|---|
| | 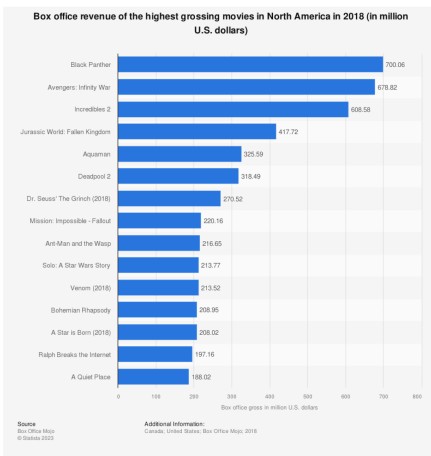 According to the image, in 2018, did "Avengers: Infinity War" achieve the highest box office revenue in North America? |
| **Question:** | |
| **Answer:** | No |
| **Qwen-2.5-VL-7B-Thinking** | <think> The image is a bar chart showing the box office revenue of the highest-grossing movies in North America in 2018. The chart lists "Avengers: Infinity War" at the top with a revenue of 678.82 million U.S. dollars. There is no movie listed with a higher revenue than "Avengers: Infinity War" in the chart. |
| | Therefore, "Avengers: Infinity War" achieved the highest box office revenue in North America in 2018. </think> Yes |
| **LLaVA-Critic-R1+** | <think> To determine if "Avengers: Infinity War" achieved the highest box office revenue in North America in 2018, I'll follow these steps: |
| | 1. Identify the movie with the highest box office revenue from the chart.  2. Compare the revenue of "Avengers: Infinity War" with the highest revenue. |
| | Step 1: The chart shows "Black Panther" with the highest box office revenue at 700.06 million U.S. dollars. |
| | Step 2: "Avengers: Infinity War" has a box office revenue of 678.82 million U.S. dollars, which is less than the revenue of "Black Panther." |
| | Therefore, "Avengers: Infinity War" did not achieve the highest box office revenue in North America in 2018. </think> No |

Table 11: An example where Qwen-2.5-VL fails in reasoning despite following the thinking template, while LLaVA-Critic-R1+ succeeds in reasoning correctly. LLaVA-Critic-R1+ accurately counts the number of figurines in the image and ultimately arrived at the correct answer

| Example 2 | |
|---|---|
| |  |
| **Question:** | How many figures are there? (Hint: include the figurine in the background) |
| **Answer:** | 10 |
| **Qwen-2.5-VL-7B-Thinking** | `<think>`
To determine the number of figures in the image, I'll count each distinct figure, including the one in the background.
1. There is a large figurine in the background.
2. There are 8 smaller figurines in the foreground.
Adding these together: 1 (background) + 8 (foreground) = 9.
`</think>` Final answer: 9 . |
| **LLaVA-Critic-R1+** | `<think>`
To determine the number of figures, I'll count each distinct figurine in the image, including the one in the background.
1. There is a large figurine in the background.
2. There are 9 smaller figurines in the foreground.
Adding these together: 1 (background) + 9 (foreground) = 10.
So, the total number of figures is 10.
`</think>` 10 |

Table 12: An example where Qwen-2.5-VL fails in grounding for GUI agent tasks on OSWorld benchmark. The `Resulting State` images show the interface after executing the proposed action. Our model follows one of the possible valid trajectories to successfully click the `Format` button, whereas Qwen-2.5-VL clicks on an incorrect location.

---

**Example 3**

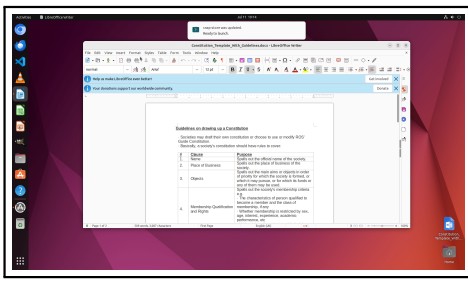

| | |
|---|---|
| **Task:** | Help me center align the heading in LibreOffice. |
| **Possible Trajectory:** | Click `Format` → Hover `Align Text` → Click `Centered` |
| **Qwen-2.5-VL-7B** | `{"action":"left_click","coordinate":[539,177]}` |
| | `Resulting State:` |

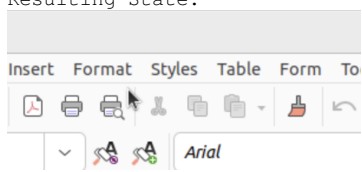

| | |
|---|---|
| **LLaVA-Critic-R1+** | `<think>` To center align the heading, I need to locate the paragraph formatting options in the toolbar. This is typically found in a section of the toolbar that includes various alignment options. Since the Format menu is likely the target, I'll click on the Format menu, expecting to find options to center align. `</think>` |
| | `{"action":"left_click","coordinate":[515,160]}` |
| | `Resulting State:` |

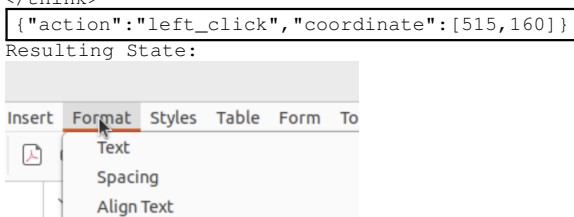

---