# OpenReview forum: "LLaVA-Critic-R1: Your Critic Model is Secretly a Strong Policy Model"
_ICLR.cc/2026/Conference — Submitted to ICLR 2026_

### Official Review · Reviewer_Tdfe · 2025-10-17

**Soundness:** 3
**Presentation:** 3
**Contribution:** 1
**Rating:** 4
**Confidence:** 4

**Summary:**

This paper proposes a method for training a multimodal model that can function as both a critic and a policy. The authors take a standard pairwise preference dataset, originally designed for training critic models via Supervised Fine-Tuning (SFT), and reformat it for a Reinforcement Learning (RL) setup. They then apply the Group Relative Policy Optimization (GRPO) algorithm to a base Vision-Language Model (VLM). The central claim is that this process, while designed to train an evaluation (critic) model, "surprisingly" also yields a strong generation (policy) model. The resulting model, LLaVA-Critic-R1, is evaluated across a wide range of benchmarks, showing some improvements over its base model. The paper also demonstrates a test-time scaling technique where the model generates multiple candidate responses and then uses its critic capability to select the best one.

**Strengths:**

Comprehensive Experimental Evaluation: The authors have conducted an extensive and thorough empirical study. The models are evaluated across a wide array of 26 different benchmarks, covering diverse capabilities such as general VQA, complex image reasoning, chart understanding, and even agentic tasks. Furthermore, the paper includes numerous ablation studies (e.g., Tables 8 and 9) that systematically investigate the effects of different training strategies and data configurations). This level of empirical rigor provides a comprehensive picture of the proposed model's performance.

Clarity of Presentation: The paper is well-written and clearly organized. The authors explain their methodology and experimental setup in a straightforward manner, and the results are presented through numerous tables and figures that are easy to follow. This clarity facilitates the reader's ability to understand and critically assess the paper's claims, even if one ultimately disagrees with their significance

**Weaknesses:**

My primary concerns with this paper are its limited novelty, marginal empirical gains from the core training method, and overstated claims.
1. Incremental Novelty: The methodological contribution of this work is exceptionally thin. The core training algorithm is GRPO, a well-established and widely used algorithm in the RL community. The training setup involves using pairwise preference data to provide a reward signal, which is the foundational concept of Reinforcement Learning from Human Feedback (RLHF).
2. Marginal Empirical Gains: The paper's core claim of producing a "strong policy model" is not convincingly supported by the results.
  - Modest Training Improvement: A close look at the tables (e.g., Table 3) reveals that the performance gains from the critic RL training itself are quite modest. The overall average for LLaVA-Critic-R1 (57.38) is only marginally better than that of a strong policy-trained model like ThinkLite-VL-7B (56.76). In many individual benchmarks, the improvements are incremental at best. The bar charts in Figure 1 visually confirm this, where the performance difference between the proposed model and the baselines is often slight.
  - Conflating Training and Inference Gains: The most impressive numbers (e.g., +16.5% and +11.1% in Table 4) come from the computationally intensive "Best-of-128" self-criticism at test time. This is an inference-time search strategy, not a direct result of the training method's superiority. The paper's presentation conflates these two sources of improvement, potentially giving a misleading impression of the core method's effectiveness.
3. Overstated "Surprising" Finding: The central narrative of the paper revolves around the "surprising" discovery that a critic model can also be a strong policy model. This finding is arguably predictable. Training a model to deeply understand and compare two detailed responses to determine which is better inherently requires it to learn the underlying principles of what constitutes a good response. It is a natural and expected outcome that such training would improve its own generative capabilities. Framing this as a surprising breakthrough appears to be an attempt to inflate the significance of the results.

**Questions:**

1. Regarding Novelty: Could you please clarify the core technical innovation of this work beyond the application of the standard GRPO algorithm to a preference dataset?
2. Regarding Performance Claims: The performance gains from the critic training phase alone seem incremental when compared to strong, specialized policy models. How do you reconcile these modest gains with the strong claim that your method produces a "secretly strong policy model"?
3. Regarding Inference Cost: The "Best-of-128" self-criticism is responsible for the most significant performance boosts. Could you provide a detailed analysis of the inference-time computational cost (e.g., latency, total FLOPs) of this method versus a single forward pass? It is crucial to understand if the performance gains justify the massive increase in compute.
4. Regarding the Source of Improvement: The ablation study in Table 5 shows that simply forcing a "think-then-answer" template on the base model already provides a noticeable boost in reasoning tasks. How can you more clearly disentangle the performance gains that come specifically from learning the preference signal, versus the gains from simply enforcing a structured reasoning format via the format reward?

---

### Official Review · Reviewer_mbHL · 2025-10-28

**Soundness:** 3
**Presentation:** 3
**Contribution:** 2
**Rating:** 4
**Confidence:** 3

**Summary:**

This paper introduces LLaVA-Critic-R1, a critic model for vision-language preference modeling. It proposed reorganizing preference-labeled critic datasets into verifiable, machine-readable reward signals and apply reinforcement learning GRPO to train a single model, LLaVA-Critic-R1. Apart from improving critic capabilities, it is shown the critic model yields a better policy model, boosting performance across visual reasoning benchmarks. Notably, leveraging self-critique at inference further amplifies performance without extra training, and ablations suggest that critic RL training enhances both structured reasoning and perceptual abilities.

**Strengths:**

- The paper is carefully written, clear and well organized.
- VLM-as-a-Judge is a timely and important topic.
- The produced models achieve sota performance on public benchmarks on their scale.

**Weaknesses:**

- The contribution seems limited. There has been quite a few studies demonstrating the effectiveness of RL for VLM reasoning [1][2][3][4]
- Some result interpretations need further justification. In Section 4, the authors mentioned "This suggests that enhancing perception capabilities also contributes to performance improvements on visual reasoning tasks." However, there is no significant evidence showing that the proposed training pipeline enhances the VLM's perception capabilities.
- It would make the paper stronger if some evaluations with regard to the critic model performance could be make on related benchmarks, for example, VLRewardBench [2]

[1] Improve Vision Language Model Chain-of-thought Reasoning
[2] R1-Reward: Training Multimodal Reward Model Through Stable Reinforcement Learning
[3] VLM-R1: A Stable and Generalizable R1-style Large Vision-Language Model
[4] InternLM-XComposer2.5-Reward: A Simple Yet Effective Multi-Modal Reward Model
[5] VL-RewardBench: A Challenging Benchmark for Vision-Language Generative Reward Models

**Questions:**

- Is the structured reasoning implemented through instruction/prompting and format reward?
- The pairwise training could introduce some noises, particularly false positives when the model has 50% probability randomly guessing the correct answer. This could lead to reward hacking or model collapse. How do you deal with this?
- Why does the critic training improves the perception of the base model? It makes sense to me that it improves the long CoT, but I think it need more evidence to show the connections to perception.

---

### Official Review · Reviewer_8nbU · 2025-11-01

**Soundness:** 3
**Presentation:** 3
**Contribution:** 2
**Rating:** 6
**Confidence:** 4

**Summary:**

This paper reorganizes preference-labeled critic datasets into verifiable RL signals and trains a unified model (LLaVA-Critic-R1) that acts both as an evaluator and a generator.
Through RL fine-tuning on critic data, the model simultaneously improves critic judgment and policy performance, achieving SOTA results on a series of reasoning/understanding benchmarks.
Moreover, the paper demonstrates self-critic test-time scaling, where the model iteratively generates and evaluates its own outputs, yielding an additional gain.

**Strengths:**

- Unified Critic-Policy Framework: Converts critic data into RL rewards, bridging evaluation and generation within a single model.
- Comprehensive evaluations: Demonstrated improvements across 26 multimodal reasoning benchmarks and multiple base architectures.
- The writing is easy to follow.

**Weaknesses:**

1. More experiments to demonstrate the test-time scalabiliy:
- Using LLaVA-Critic-R1 as an external critic for other models;
- Testing the impact of different critic models (llava-critic) or prompting templates (please refine ...) at inference.
2. Novelty ambiguity: While combining critic training with RL is elegant, similar “reflection” or “self-evaluation” paradigms exist in recent RL-based VLMs [1][2] or actor-critic architectures [3]. The paper should better clarify conceptual differences and necessity in related works.
3. Effectiveness for critic reward: Why does critic training lead to improvements in both perception and reasoning? The paper claims this is due to the critic-reward. If one trains via RL on two different subsets of data that use the same image, but one subset is a critic-task and the other is a standard QA task (i.e., just generating the answer), would we expect the same effect in perception and reasoning performance?

[1] Wang et al. VL-Rethinker: Incentivizing Self-Reflection of Vision-Language Models with Reinforcement Learning. arXiv:2504.08837\
[2] Chen et al. SFT or RL? An Early Investigation into Training R1-Like Reasoning Large Vision-Language Models. arXiv:2504.11468\
[3] Wei et al. Perception in Reflection. arXiv:2504.07165

**Questions:**

- Please see weakness.
- Optimization conflict discussion: whether combining critic and actor's supervision leads to gradient interference or performance conflicts? SFT-based models have shown that joint training on both critic-style evaluation tasks and normal vl generation tasks can degrade overall performance due to conflicting optimization signals. It would be valuable to know whether the RL formulation here encounters similar issues

---

### Official Review · Reviewer_JZQx · 2025-11-07

**Soundness:** 3
**Presentation:** 3
**Contribution:** 3
**Rating:** 4
**Confidence:** 4

**Summary:**

This work creates LLaVA-Critic-R1 to fix the split between VLMs’ critic (only scores) and policy (only generates) roles. It turns preference-labeled critic data into RL-ready signals, fine-tunes a base model (Qwen-2.5-VL-7B) with RL (using preference+format rewards, GRPO optimization), and gets a unified model good at both evaluating and generating.
It beats the base model by 5.7% on 26 visual benchmarks, matches/surpasses specialized reasoning VLMs; the upgraded LLaVA-Critic-R1+ hits SOTA 71.9 on MMMU (7B scale); test-time self-critique adds 13.8% on reasoning tasks (no extra training). The authors also prove it works via better visual perception and structured reasoning.

**Strengths:**

1. Novel approach: Using RL to build a unified model, and showing that training a critic can also make the policy better.
2. The experiments are well-designed: tested on 26 benchmarks (covering perception, reasoning, chart understanding, video tasks), validated on multiple base models (Qwen, MiMo-VL, Llama-3.2-Vision), and did ablation studies (like different training orders, SFT vs. RL). The results are reliable and can be applied to other models.
3. It solves the problem where traditional critic models "only copy external judgments"—they removed GPT-generated rationales during training, forcing the model to reason on its own. That makes its judgments more independent.

**Weaknesses:**

1. It doesn’t clearly explain why policy performance drops later in training. They mention overfitting, but no details—like which tasks or data cause it, or if it’s over-adapting to critic data and hurting general generation.
2. No testing with smaller datasets: they used 40k critic data points the whole time, never tried 10k or 20k. If it fails with less data, it’ll be less useful for real-world deployment.
3. No analysis of self-critique failures: For example, LLaVA-Critic-R1+ is better at generation but worse at self-critique. No explanation—whether it’s bad evaluation standards or not enough diversity in candidate responses. No targeted troubleshooting here.

**Questions:**

Please refer to the Weaknesses section above for details.

---

### Meta-Review · Area_Chair_RxDV · 2025-12-08

**Summary:**

Four knowledgeable reviewers reviewed this paper and raised the following concerns:
1. Validation is not fully convincing:
    - Experiments are limited and do not consider smaller dataset sizes (JZQx) nor benchmarks such as VLRewardBench (mbHL)
    - Additional experiments would be needed to demonstrate test-time scalability and generalizability to different critics or prompting (8nbU)
    - Some claims are not properly supported with experimental evidence (mbHL, Tdfe)
    - The significance of results remains unclear (Tdfe)
2. No analysis of failure modes (JZQx), nor intuitions on why the critic training leads to improvements (8nbU)
3. The novelty remains unclear (8nbU, mbHL, Tdfe), clarifying the positioning of the work w.r.t. prior art would be beneficial

**Reviewer Concerns:**

There was no rebuttal so all reviewers' concerns remain.

**Reviewer Scores:**

There was no rebuttal so no effect on scores/discussion.

---

### Decision · Program_Chairs · 2026-01-26

Reject